# Mechanical Characterisation of GFRP Frame and Beam-to-Column Joints Including Steel Plate Fastened Connections

**DOI:** 10.3390/ma15238282

**Published:** 2022-11-22

**Authors:** Giuseppe Ferrara, Olivier Helson, Laurent Michel, Emmanuel Ferrier

**Affiliations:** 1Laboratory of Composite Materials for Construction (LMC2), University Claude Bernard Lyon 1, 82 bd Niels Bohr, CEDEX, 69622 Lyon, France; 2Departement Fluides, Thermique, Combustion, UPR 3346 CNRS, Institut Pprime, University of Poitiers, ENSMA, BP 40109, 86961 Poitiers, France

**Keywords:** composite structure, beam-to-column connection, pultruded fibre-reinforced polymer (FRP), portal frame joints, MOOVABAT project

## Abstract

The study is part of the MOOVABAT project aiming at defining innovative technological buildings with low environmental impact and characterised by the capacity to constantly adapt to the changing of their users’ needs. In this context, the mechanical performance of a fibre-reinforced polymer (FRP) frame, chosen as a structural solution for the building assembly, was investigated. Specifically, the research study aims to experimentally define the moment–rotation behaviour of screw-connected joints by using steel plates. For this purpose, two different configurations, a beam-to-column joint and a whole portal frame, were tested to evaluate the strength and the stiffness of the connection. In addition, the beam-to-column element was also subjected to cyclic loads to assess the joint energy dissipation capacity. The experimental results show that the strength of the connection is higher than that required to satisfy both serviceability limit state (SLS) and ultimate limit state (ULS) loading conditions. Moreover, it also provided an accurate characterisation of the semi-rigid connection useful for designing purposes and raising the possibility of considering an optimisation of the system. All in all, with respect to mechanical aspects, the study confirms the suitability of pultrude FRP element assemblies for modular building applications and paves the way for further analysis aimed at enhancing their efficiency.

## 1. Introduction

Buildings for high-tech industries are often subjected to significant changes in their structure to meet new requirements due to a constant evolution of the users’ needs [1]. These structural modifications may result in significant economic and environmental impacts. In this context, the MOOVABAT project, joining the expertise of companies in the field of energy and construction, and research laboratories, was conceived with the aim of defining innovative technological building characterised by a low environmental impact, and by an adequate versatility to promptly respond to the demands of this dynamic and constantly evolving context. The use of a modular building, generally requiring a few weeks for both designing and construction, and easily adaptable to several configurations, emerged as a smart technical solution for this purpose [2,3].

Several construction solutions, guaranteeing fast implementation, versatility in their application and satisfying mechanical properties, can be considered for the structure of modular buildings [4,5,6,7]. Notably, wood–concrete composite structures, assemblies of pultruded fibre-reinforced polymer (FRP) elements, and precast concrete emerged as valuable options [8,9,10].

In the last decades, pultruded FRP composite systems gained increasing attention due to very attractive mechanical properties including a highly elevated specific strength (strength-to-weight ratio), high elastic modulus, easy-to-assemble, good resistance to chemical aggression, and design flexibility [11].

FRP composite systems are structural engineering materials comprising high-strength continuous fibres, of about 10 μm diameter, embedded in a continuous polymeric matrix [12]. Carbon and E-glass synthetic fibres are the most employed reinforcement types for the implementation of the so-called CFRP and GFRP composites for structural engineering applications. Unsaturated polyester, vinyl ester and epoxy resins are the principal substances used for commercialised FRP systems. Typical GFRP profiles include E-glass continuous filament mats or woven fabrics, providing transversal strength, and unidirectional fibre threads running along the pultrusion length, providing longitudinal strength and stiffness.

Several research studies were carried out to characterise the structural behaviour of pultruded FRP elements, considering different sections and structural members (such as beams, columns, and bridge decks), showing that they are suitable for applications in civil engineering [13,14]. In this context, pultruded FRP frames emerged as a valuable solution for sustainable prefabricated houses due to satisfying mechanical performance and representing an eco-friendlier alternative to traditional steel and concrete structures, known as very polluting materials [15].

The intense research activity of the scientific community in this field, although emphasising the great potential of the structural application of FRP pultruded members, and defining their mechanical performance, it also highlighted several aspects on which it is necessary to pursue studies to have exhaustive designing criteria. Notably, further research is needed to assess the impact of environmental conditions on the long terms structural behaviour, to define the dynamic response, to accurately describe buckling phenomena, and to investigate issues related to members connection systems [16].

Due to the increasing market volume of FRP applications as structural elements, it emerged the need of developing standardization documents to support both manufacturers and practitioners with practical rules for the design and verification of FRP structures in accordance with safety and serviceability requirements. A prospect of forthcoming guidelines and rules for structural analysis and design of FRP used in load-bearing structures was edited in 2016 to stimulate the debate on the future standard [17]. This document, although not the official reference, represents a valuable support for the design of FRP structures. The lack of regulatory framework encourages testing procedures assisting the design process to increase the reliability of FRP assemblies in civil engineering applications. For this reason, several studies were carried out to experimentally characterise joints between FRP structural elements, including bolted or riveted solutions comprising FRP or steel connecting elements [18,19,20]. These studies, combined with numerical applications, emphasised the importance of an accurate definition of the moment–rotation properties of the joint in order to have a reliable prediction of the frame’s mechanical behaviour [21,22]. It also highlighted the need of comparing the mechanical performance of FRP structural elements with serviceability and ultimate limit states to show the suitability of the material for designing purposes [23,24].

In this context, the present study aims at investigating the mechanical performance of GFRP assemblies in view of their application as modular structures in the framework of the MOOVABAT project. Specifically, the moment–rotation of beam-to-column joints, comprising fastened steel gussets plates, is investigated both in monotonic and cyclic load conditions in order to characterise strength and dissipative properties. In addition, a whole FRP frame was tested, by applying in-plane lateral forces in order to characterise the response of the structure focusing on the behaviour of the joints and of the pultruded FRP-connected members.

The results show that the system is largely capable of withstanding the required loading configurations to satisfy both SLS and ULS conditions. However, by showing a shear failure of the screws connecting the steel gussets to the FRP members, it emphasises the possibility of an optimisation of the system efficiency with a higher exploitation of the connected elements. To this purpose, by providing the actual response of the semi-rigid joints, the research study paves the way for accurate numerical analysis aimed at enhancing the efficiency of the structure, perhaps by reducing the FRP pultruded profile sections. In addition, it emphasises the importance of a proper definition of the moment–rotation behaviour of the joints in numerical analysis when the hypothesis of fully rigid connections is not reproduced in the actual structure.

The paper includes a first part in which the MOOVABAT prototype is described, and a preliminary structural analysis is conducted to define the magnitude of the expected actions. Then, the material adopted, with specific attention to the joint architecture, and the methods of the experimental activity are outlined. The results are presented and discussed with the definition of the semi-rigid behaviour of the frame joints. Finally, a simplified numerical application is reported to show the influence of semi-rigid joint modelling in the prediction of the frame mechanical response.

## 2. Design Details of the MOOVABAT Modular Element

### 2.1. Description of the System

Figure 1 shows the structure of the MOOVABAT modular element, representing the elementary part of the technological building. It consists of two longitudinal frames connected by two transversal frames. Both columns and beams are pultruded GFRP elements with a closed rectangular section, well suited for the box shape of the modular element. The joints are characterised by rigid steel plates fastened to the pultruded GFRP members. This type of connection, ease to be assembled and disassembled, is well suited for application in modular elements prone to adjustment interventions.

The facades and the horizontal elements are implemented by adopting the “Wall E+^®®^ construction system”, innovative building system made with pultruded GFRP blocks, specifically designed to be adapted to the different required geometry allowing a wide range of possibilities in the openings setting, characterised by excellent thermal and acoustic insolation properties and conceived in line with sustainability principles [21]. Concrete is poured onto the prefabricated elements of the deck in order to obtain a rigid slab and to reduce the vibrations of the system.

Secondary elements, partition walls and suspended celling, consists in prefabricated panels with high thermal and acoustic insulation capacity, and are easily adaptable to different required geometries.

### 2.2. Preliminary Structural Analysis

The modular element shape and weight are constrained by the capacity of the means of transportation. Hence, the maximum weight load of the structure is superiorly limited by 70 kN. During the operational phase, the modular elements are simply supported by two linear supports 40 mm large, along its shorter edges comparable to the support width used in steel construction.

Beams and columns are defined in line with the modular element geometry, and according to simplified preliminary structural design schemes. Figure 2 represents the pultruded GFRP cross-section of the structural members and reports their geometry details and moment of inertia concerning the strong axis.

A simplified structural model was considered to define the magnitude of the design loads applied to the structural elements, i.e., columns and beams, and to the joints. Specifically, the structural scheme consists of a rectangular frame, simply supported at the lower corners, having columns 3.2 m high and beams 6.6 m long. Loads carried by the deck and the facades are assumed to be equally distributed on the two longitudinal frames. Table 1 lists the different loads, and their entity, considered in the analysis.

As for the installation phase, the structure, with an overall dead load equal to 45 kN, complies with the transportation constraints.

As for the SLS requirements the design deflection, imposed equal to 1/500 of the beam span, has a value of 13 mm. In line with the prospect for new guidance in the design of FRP [16] the material partial factor (*γ_w_*) for the assessment of the deflection is assumed equal to 1. In absence of information concerning the rotational stiffness of the joints, for safety reasons, the actual deflection is assessed by considering the top beam simply supported, assuming the column-to-beam joint as a hinge [25]. With this assumption, the deflection at mid-span is equal to 4.3 mm. According to the design guidance [16], this value is divided by a total conversion factor, *η_c_*, that considers the effects of temperature, humidity, creep and fatigue (Equation (1)).
(1)$$\eta_{c}=\eta_{ct}\cdot \eta_{cm}\cdot \eta_{cv}\cdot \eta_{cf}$$

In the equation *η_ct_* is the conversion factor for temperature effects assumed equal to 0.90, *η_cm_* is the conversion factor for humidity effects assumed equal to 0.70, *η_cv_* is the conversion factor for creep effects assumed equal to 0.62, *η_cf_* is the conversion factor for fatigue effects assumed equal to 1. The total conversion factor has a value of 0.39. The actual deflection results equal to 11 mm and complies with the SLS requirement [26].

As for the ULS without detailed information on the moment–rotation behaviour of the joints, within the preliminary analysis, they were assumed as fully restrained. This assumption, although not properly representative of the type of connections adopted in the joint, is adopted for preliminary design purposes. It is within the aims of the study that the experimental assessment of the joint’s stiffness is to be adopted in the definition of a more realistic numerical model. In line with the standards concerning the structural design [27] several load cases were considered. The structural analysis results are summarised in Table 2 in which, for each load case the maximum value of the internal forces is reported: notably axial force (*N*), shear force (*V*), and bending moment (*M_b_*), attained in column or beams members, and bending moment attained in correspondence of the joints (*M_j_*).

## 3. Materials and Methods

### 3.1. Materials

Pultruded GFRP structural members used as beams and columns within the study are characterised by a closed cross-section (Figure 2). The elements, obtained by embedding glass fibres in a polyester matrix, were provided by the company TopGlass [28], and they are labelled as TRIGLASS^®®^ profiles. The section was created by assembling U profiles constituting the bottom and the top part (60 mm high, 200 mm wide and 10 mm thick) and flat profiles placed on the sides (30.8 mm wide and 7 mm thick). The elements were assembled by using a polyurethane adhesive, with the commercial name of ADEKIT^®®^ H6236, Sika France [29], after having accurately prepared the joining surfaces with isopropanol, Wurth, France. The adhesively bonded connection is characterised by a delamination strength of 8 MPa assessed in accordance with the EN ISO 10365 standard [30].

The main mechanical of pultruded GFRP members, provided by the manufacturer, are listed in Table 3 together with the reference standard adopted for the characterisation. Properties concern both longitudinal and transversal directions.

Joints were implemented by means of steel plates fastened to the pultruded GFRP members by means of self-drilling screws. This solution was preferred to others, i.e., GFRP plate-to-plate, riveted or bonded connections, for its suitability to be easily either implemented or disassembled in view of applications in modular structures such as MOOVABAT ones.

Metallic plates have a thickness of 6.2 mm, so to behave as a rigid element uniformly distributing stresses among the screws. With respect to the prototype described in Section 2, a typical joint of the longitudinal frame includes a column and a beam connected on the lateral side by an L-shaped plate. Within the experimental activity, the influence of the transversal frames is neglected. In view of possible applications in which the modular elements are sequentially arranged, intermediate joints have to connect the column with two consecutive beams. In order to consider this configuration, this type of joint also was considered within the experimental activity. In such a case, GFRP members were connected by T-shaped steel plates.

The geometry of the joints, including the size of the plates and the screw distribution, are represented in Figure 3. Joints connecting the column with a beam are labelled as L-joints, and joints connecting the column with two consecutive beams are labelled as T-joints.

In both configurations, the geometry of the plate was defined so as to overlap the GFRP elements for a length equal to their height. L-shape and T-shape plates respectively have a weight of 21.0 kg and 36.6 kg.

Stainless steel screws, with a diameter (d_s_) of 5.5 mm, were adopted as connecting elements. Each screw had a tensile capacity of 17.00 kN and a shear capacity of 12.27 kN. A tightening torque comprised in the range of 9 to 11 Nm was adopted. This value was set to avoid the local damage of the GFRP elements characterised by a minimum thickness equal to 7 mm [27].

Fasteners were uniformly distributed along the plates’ edges. In addition, a row was added along the mid-section. Screws were spaced at a distance of at least 66 mm (12 d_s_). According to the preliminary design formula, this configuration should prevent local failures of the pultruded profiles (Figure 3).

With respect to the T-joint, a preliminary identification of the maximum capacity was carried out assuming a failure mode involving fasteners sharing. To this purpose, the steel plate was assumed stiff enough to guarantee rigid displacements of the screws. With this assumption, the shear contribution of the generic screw, *R_i_*, can be defined by means of Equation (2):(2)$$R_{i}=\sqrt{{\left(\frac{V}{n}+\frac{M\times x_{i}}{\sum^{\;}r_{i}^{2}\;}\right)}^{2}+{\left(\frac{M\times y_{i}}{\sum^{\;}r_{i}^{2}\;}\right)}^{2}}$$
where *V* and *M,* respectively, represent the shear and bending moment applied on the beam edge of the joint, *n* is the number of screws, *x_i_* and *y_i_* represent the coordinates of the generic screw with respect to the steel plate barycentre, *r_i_* is defined in Equation (3).
(3)$$\sum^{\;}r_{i}^{2}=\sum^{\;}x_{i}^{2}+\sum^{\;}y_{i}^{2}$$

In line with the configuration adopted in the experimental campaign of the study the bending moment *M* is assumed as the product of the shear force *V* and a lever arm is assumed equal to 1.2 m. Based on Equation (2), a shear force *V* equal to 60 kN is required to promote the shearing failure of the most severely stressed fastener. This design capacity results to be much higher than the estimated actions expected for the structure, reported in Table 2.

It is worth emphasising that the connection elements were chosen according to geometric and logistic reasons. Both preliminary identifications of the structure’s expected actions and joint capacity, carried out by adopting simplified methods, were aimed at ensuring that the system is at least capable of complying with SLS and ULS requirements. It is the purpose of the study to mechanically characterise the system by identifying its actual response, including joint moment–rotation stiffness and strength, to be used for the implementation of accurate models aimed at improving the efficiency of the structure.

### 3.2. Test Configurations and Procedures

The experimental study involved structural elements comprising the described GFRP pultruded beams and columns, T-shaped and L-shaped joints. Two different configurations were considered:beam-to-column: it represents a joint connecting a column to two consecutive beams by means of the T-shape connecting system (Figure 4a). Two specimens differing in the fastening elements were tested in this configuration: the first one, named T1 characterised by self-drilling screws without washers, and T2_W characterised by self-drilling screws equipped with screws;portal frame: it represents the main longitudinal frame of the MOOVABAT modular structure, characterised by two GFRP columns and two GFRP beams, connected by L-shaped joints (Figure 4b). One specimen, named PF, was tested in this configuration.

In both systems, a horizontal load was laterally applied on the column in order to investigate the response of the frames and their parts, namely the joints, in the plane.

Beam-to-column specimens were subjected to both static and cyclic loading conditions. Cyclic loading conditions comprised positive and negative displacements. The portal frame was tested in static loading conditions. The objective is to analyse the boundary conditions of the beam–columns joint and to conclude on rigid or semi-rigid connection.

The horizontal load was applied by means of a hydraulic press with a maximum load capacity of 500 kN and a stroke of 100 mm. The loading system is supported by a reaction wall guarantying the imposed load to be properly transferred to the tested structure.

#### 3.2.1. Beam-to-Column Element

The test set-up concerning the beam-to-column elements is represented in Figure 5. The joint, as well as the pultruded element section, are reproduced in full scale. Steel T-shaped plates were symmetrically placed on both sides of the joint. The lever arm of the horizontal load, with respect to the centre of the joint (identified as the intersection between the axis of the pultruded GFRP members), was equal to 1.2 m. The two edges of the beams were equipped with steel fixation elements to prevent both horizontal and vertical displacements. In proximity to the column edge, where the horizontal load was applied, a hinge was reproduced. The lengths of the GFRP members converging in the joint were defined in order to simulate the conditions to which the connecting element is subjected within the actual frame.

The two beam-to-column specimens, namely T1 and T2_W, were subjected to cyclic loading (cyclic test), and then to the final last static loading conditions with the load monotonically increasing up to the failure (static test). The cyclic test is stopped when enough damage is identified in the structure. Both cyclic and static tests were carried out in displacement control. The loading history of the cyclic test includes different steps characterised by three cycles. The first 25 steps corresponded to a displacement increase of 0.8 mm, and the next steps (5 steps for T1 and 3 steps for T2_W) corresponded to a displacement increase of 4 mm. The cyclic test was carried out with a stroke rate of 100 mm/min.

After the cyclic test, the static test was monotonically carried out up to the failure of the specimen. The static test was carried out with a stroke rate of 4 mm/min.

Two linear variable displacement transducers (LVDT) were placed on the column to monitor the horizontal displacement of the members at two different heights (LVDT 3 and LVDT 4). These displacement values were adopted to have an approximated estimation of the joint rotation during the test. Two transducers (LVDT 1 and LVDT 2) were placed on the beams to monitor their vertical displacements during the test. Three strain gauges (SG 1, SG 2 and SG 3) were placed on the flange of the column at different heights in order to have the strain distribution along the member length during the test (Figure 5).

#### 3.2.2. Portal Frame

The specimen tested reproduced the longitudinal portal frame of the modular MOOVABAT prototype at full scale. The test-set up is represented in Figure 6. GFRP members were connected by means of L-shaped joints. In each joint, the L-shaped plate was placed only on one side, corresponding to the external side of the modular structure.

No connecting elements were placed on the inner side, and the contribution of the transversal frames was neglected. Figure 7 represents a close-up of both the external and the inner sides of the connection system at the intersection between a beam and a column of the portal frame. The frame had a length of 6.6 m and a height of 3.2 m. The frame is simply supported in proximity to the lower joints (joint A and joint D). The vertical displacement of the lower beam in proximity to the joints was prevented by fixation elements, as well as the horizontal displacement in correspondence with joint D. In the service configuration, the lateral displacement of the beam’s out-of-frame plane is prevented by the slabs of the deck. In order to reproduce this condition, wood elements were transversally placed at the mid-span of the lower beam to avoid lateral transversal buckling phenomena.

The load was applied at a height equal to 2.27 m with respect to the centre of the lower joints, corresponding to 70% of the column length.

The test was carried out in displacement control with a stroke rate of 2.2 mm/s. Along the L-column, at different heights, LVDT were placed (LVDT 1, LVDT 2, LVDT 3, LVDT 4) to record the horizontal displacement distribution of the member during the test. In addition, a laser sensor was placed to record the horizontal displacement of the column in proximity to the load application point. Strain gauges sensors were applied over the web of the L-column, in correspondence with two different heights, to have the flexural deformation of the pultruded element (six strain gauges at each height). Four strain gauges were placed at the external flange of the L-column. Two LVDTs (LVDT 5 and LVDT 6) were respectively placed in correspondence with the L-beam and the T-beam to monitor the vertical displacement of the members during the test. The position of each measurement device is reported in detail in Figure 6.

## 4. Results and Discussion

### 4.1. Beam-to-Column Element

The mechanical response of beam-to-column elements subjected to a horizontal force applied on the vertical member is presented first in terms of load and horizontal displacement of the top edge of the column. This representation of the results allows consideration within the analysis of local displacements of the connecting elements, and energetic aspects. In addition, the results can be easily compared with a simplified numerical model reproducing the tested element. Secondly, the results are presented in terms of moment–rotation response. This representation, typically adopted for similar systems [31], allows to describe the mechanical response of the system regardless of the adopted geometry, and so to make possible a classification of the joint in terms of rotational stiffness, in line with standard criteria [32].

#### 4.1.1. Load-Displacement Response

Figure 8 shows the hysteresis load–displacement curves concerning beam-to-column elements T1 and T2_W tested in cyclic loading conditions.

Each cycle, in line with typical hysteresis curves, comprised a pushing loading and unloading phase and a pulling loading and unloading phase. The displacements, obtained by means of LVDT 4, were negative in the pushing phase, and positive in the pulling phase.

Both specimens T1 and T2_W showed a hysteresis curve symmetrical with respect to the origin, and each cycle was rather well-centred at the origin. Therefore, the system exhibited a similar response in both pulling and pushing conditions.

Figure 8 also reports the envelope of the curves concerning the different cycles, so as to have a comprehensive identification of the response of the system under cycling conditions. The envelope was characterised by a fist steeper branch up to a load of about 6 kN. The stiffness slightly decreased in the following cycles and remained nearly constant until the end of the test. The stiffer response at lower load values is due to the contribution of the friction between the elements of the connection, i.e., GFRP members and steel plate. The magnitude of the frictional contribution is related to the screws tightening torque adopted. Once the applied load exceeded the frictional contribution, a relative displacement between the GFRP members and the steel plate occurred due to the fastener’s holes tolerance.

During the loading phase, the system exhibited both elastic and sliding deformation. Due to local damaged deformation, the stiffness in the unloading phase resulted lower. In fact, due to this sliding phenomenon, the energy conferred to the system in the loading phase was partially dissipated in the unloading phase. As consequence, after a complete unloading phase, the system exhibited a residual displacement. Figure 9 shows the residual displacement as a function of the applied load achieved in the different cycles, for both specimens T1 and T2_W. The curves show three phases: the first phase is characterised by small displacement, the second phase is characterised by a significant increase in the displacements, and the third phase is characterised by a quasi-constant displacement value. This representation confirmed that up to a load value of about 6 kN, in the first phase, the system exhibited an elastic response, without significant residual displacements. For higher load values, the loss of frictional capacity due to a slippage between the steel plate and GFRP members resulted in a logarithmic increase of the residual displacement up to a load value of about 30 kN. For higher load values, in the third phase, the residual displacement was roughly constant and included in the range between 5 mm and 7 mm. It can be noticed, in the second phase, that being equal to the load value, specimen T1 is characterised by higher values of the residual displacement with respect to the T2_W specimen, and it increases with the load following a less steep trend. This result emphasises that the sudden increase of residual displacement after the achievement of the frictional strength, is delayed by the presence of the washers. For specimen T2_W, the small decrease in displacement (−0.5 mm) after the load of 50 kN should not be considered, this is certainly due to the small displacement in the testing device. This result strengthened the mechanical behaviour and make it comparable to timber connection systems with a first increase in sliding and then a stabilization.

Each hysteresis loop, either in the pushing or pulling phase, comprised a loading and an unloading branch. In the unloading branch, part of the energy was dissipated by the system. For each phase, the energy stored was assessed as the area underlying the unloading branch in the load–displacement curve, while the energy dissipated was defined as the difference between the area underlying the loading branch and the stored energy. Figure 10 shows the energy stored and the energy dissipated in each cycle as a function of the load (maximum load achieved by the cycle) for both specimens T1 and T2_W. For both specimens, the stored energy increased as the squared of the load. The dissipated energy attained neglectable values in the first branch, confirming the elastic behaviour of the system before the slippage between the joint components, then it increased following a quasi-linear trend up to a load of about 45 kN. The dissipation of energy in this range may be due to the rotation of the screws in the holes, which is proportional to the applied load, hence the consequent displacement. For higher values of the load the dissipated energy increased following an exponential trend. This sudden increase in dissipated energy was due to the plasticisation and failure of most stressed fasteners.

Figure 11 shows the load–displacement response of T1 and T2_W specimens tested in the last static loading conditions until failure, after the cyclic tests were completed. The specimens exhibited an initial displacement, comprised in the range between 5 mm and 7 mm, corresponding to low values of the load. This displacement corresponded to the final residual displacement deriving from the cyclic loading tests (Figure 9). For higher values of displacement, the specimens exhibited a quasi-linear response up to the onset of the failure. The stiffness of the connection in this linear part is equal to 2.5 kN/mm. It can be comparable to the stiffness of steel-to-timber screw connections according to De Santis et al. [33] and M. Schweigler [34]. The latter occurred with the progressive failure in shear of the most stressed screws. The specimen T2_W, due to the presence of the washers, exhibited the beginning of the linear phase at lower values of displacement. Both the specimens were characterised by a similar maximum loading capacity, being respectively equal to 70.1 kN and 67.2 kN T1 and T2_W beam-to-column element. These values are much higher than the shear values expected from structural analysis (Table 2).

The experimental response was compared to a simplified numerical model reproducing the test conditions and assuming the joint with a rigid behaviour. The structural scheme adopted for the mechanical analysis is represented in Figure 12, including geometric details and the rigidity adopted. A usual beam model is used to get a force–displacement response. For both column and beam members, the moment of inertia was assessed with respect to the cross-section presented in Figure 2, and the Young’s modulus was assumed as the tensile module of elasticity in the longitudinal direction (Table 3). A monotonic increase in the load was simulated and the corresponding horizontal displacement at the head of the column was considered output data. All the calculations are performed with an elastic linear hypothesis of materials and boundary conditions. The load–displacement response deriving from the numerical analysis is represented as well in Figure 11. The resulting behaviour may be considered the theoretical behaviour of the connection, in linear conditions, and with a rigid joint. The aim of this is in a first approach to identify the boundary conditions of the connection. Further deeper mechanical modelling could be performed to include the slipping effect of the connection which is not the aim at this stage but FEM or non-linear analysis may be performed [35,36]. The numerical curve showed a slope higher than that one exhibited by the experimental ones. This aspect emphasised that the interaction between the connecting elements, i.e., rotation of the screws and slipping between steel plate and GFRP members, led to an overall response in which the beam-to-column connection behaves as a semi-rigid joint.

#### 4.1.2. Moment-Rotation Response

The bending moment was assessed as the product between the applied horizontal load and the lever arm with respect to the geometrical barycentre of the joint. In absence of specific equipment for the recording of the joint rotation, a simplified estimation was made by using the displacement transducers adopted. Specifically, the displacement recorded by LVDT 1, LVDT 2 and LVDT 3 was considered (Figure 5). For each member, column and beam, the rotational angle was assumed as the angle needed to generate the LVDT displacement, considering a rigid rotation of the element in the portion comprised between the joint centre and the displacement transducer. The joint rotation was assessed as the mean of the three values respectively identified on the beams and the column. Figure 13 shows the response in terms of moment–rotation of T1 and T2_W specimens tested in static loading conditions. The rotational stiffness, assessed as the slope of the moment–rotation curve in the linear branch included in the range between 20% and 80% of the maximum moment capacity, was equal to 2.71 MNm/rad and 2.5 MNm/rad, respectively, for T1 and T2_W specimens. A mean value equal to 2.6 MNm/rad can be taken for further analysis on the frame. The joint can be classified, in terms of rigidity, according to the EN 1993-1-8:2005 [32]. Notably, rigidity is a function of flexural stiffness (EI/L) and can classify the joint as rigid, semi-rigid or hinge.

According to EN 1993-1-8:2005 [32], the limit is given if the flexural stiffness is lower than 0.5 EI/L (hinge behaviour Figure 13). According to this classification, the connection system studied is classified as a joint with a semi-rigid rotational behaviour even if the behaviour is nearby to be a hinges behaviour. This aspect is in line with what is observed in Figure 11 where the specimens T1 and T2_W show an experimental load–displacement response with a lower stiffness with respect to the numerical behaviour assessed by assuming a rigid rotational behaviour for the joint.

As matter of fact, the joint did not exhibit a fully rigid behaviour due to the plastic deformations of the connection elements, i.e., the rigid inclination of the screws within the holes, and their plasticisation, as shown by the failure mode observed. Therefore, the deformability of the entire beam-to-column system was mainly concentrated in the joint. This aspect is also emphasised by analysing the strain distribution along the column. Figure 14 shows the strain value recorded by the different strain gauges placed on the column flange (Figure 5) at different representative values of the load in both the pushing and pulling phases. With respect to the column portion out of the connection, comprised between the load application point and the T-shape steel plate, the strain distribution is quite linear and monotonically increases with the load. In addition, no significant discrepancies in the strain distribution are observed between the pushing and pulling phases. It can be assumed that the column deformation was proportional to the bending moment distribution and remained in elastic conditions until the failure. On the contrary, the strain distribution concerning the portion of the column connected to the T-shape steel plate (strain gauges 1 and 2) did not follow the liner distribution observed along the column. Specifically, despite a linear increase in the bending moment up to the joint centre, the strain values were lower with respect to theoretical values following the linear trend. This experimental evidence is due to a higher flexural stiffness of the cross-section in the connected zone, due to the contribution of the steel plate, characterised by a higher module of elasticity, and increasing the moment of inertia of the system. Figure 14 also reports the moment of inertia values concerning the connected zone and the portion out of the connection of the column. In addition, the strain distribution concerning strain gauge 1 did not follow a monotonic increase up to the failure, but a reduction of the strain was observed by increasing the load at some steps of the test. This aspect, emphasising the non-elastic response of the system in correspondence with the steel plate, confirms that the connecting elements were characterised by plastic deformations.

The maximum moment capacity attained during the monotonic tests was equal to 82 kNm and 80 kNm, respectively, for T1 and T2_W specimens. By comparing these values with the expected moment required to satisfy SLS and ULS limitations, equal to 6.6 kNm (Table 2), the system capacity results to be about 12 times higher than the demand. This aspect, although emphasising that the system largely respects the standard requirements both at the SLS and ULS, also highlighted that it resulted in oversized with respect to the expected demand. As discussed in the Materials and Methods section the joint, although respecting preliminary design conditions, was mainly designed to guarantee practical logistic aspects related to the geometry of the prefabricated elements adopted in the modular prototype, and to the installation/disassembly processes. However, the experimental evidence shows that it is possible to optimise the number of screws of the connecting system in order to reduce oversizing issues.

### 4.2. GFRP Portal Frame

#### 4.2.1. Load-Displacement Response

The mechanical response of the GFRP frame subjected to a monotonic in-plane lateral load is expressed in terms of the load–displacement curve. The load represents the value of the charge applied on the left column; the displacement is the value recorded by the Laser (Figure 6). The response deriving from the experimental analysis is represented in Figure 15. The response was characterised by a pseudo-linear behaviour up to the achievement of the maximum load, equal to 45.3 kN, then a plastic behaviour was in correspondence to the failure.

With the aim of comparing the GFRP frame capacity with the required loads expected (Table 2) a simplified distribution of the forces through the structural members was considered. By assuming an equal distribution of the horizontal load between the two vertical members, each column is characterised by a constant value of the shear equal to half of the applied load (Equation (4)). The moment in the joint can be assumed as the product between the shear value and the distance between the horizontal applied load and the barycentre of the lower joint, d* (Equation (5)).
(4)$$V_{R}=\frac{F}{2}=22.6\;\mathrm{kN}$$
(5)$$M_{j}=\frac{F}{2}\cdot d^{*}=51.3\;\mathrm{kNm}$$

Both shear and bending moment resistant values are significantly higher than the values expected to act on the structure to comply with both SLS and ULS requirements (Table 2). This aspect, although highlighting that the system has sufficient structural capacity, also emphasises oversizing issues.

The failure mode observed comprised the progressive failure in shear of the screws within the joints. The joints on the right side (joint C and joint D) were characterised by more significant damage. This aspect may be attributed to the test-set up, notably to the presence of a steel block placed in proximity to joint D to prevent horizontal displacements that may have triggered an asymmetry in the frame’s mechanical behaviour, with a concentration of the stress on the right side.

The failure mode observed in the GFRP portal frame, characterised by a rupture of the connecting elements without significant damage in the GFRP structural members, was consistent with the behaviour shown by the beam-to-column elements. The moment capacity of the joints of the frame resulted in about a third lower than the capacity exhibited by the beam-to-column element. This aspect is due to the different configurations of the joints, notably the joints of the beam-to-column specimens were characterised by T-shape steel plates on both sides, while the joints of the portal frame were connected on one side. Moreover, internal joints are generally more resistant to external joints connecting a lower number of structural elements [37,38].

As matter of fact, the analysed structural system, although emphasising oversizing issues, showed a mechanical behaviour characterised by a concentration of plastic deformation in the connecting elements. The localisation of the damages in the connecting elements, without affecting beams and columns, may represent an important aspect in view of repairing interventions in which the structural members may be reused.

#### 4.2.2. Comparison with Numerical Analysis

A simplified numerical model reproducing the conditions of the experimental test was implemented to describe the mechanical behaviour of the GFRP portal frame. Figure 16 shows the structural scheme adopted for the modelling.

The cross-section of the GFRP members was defined in consistence with Figure 2, in order to have the same moment of inertia. The Young’s modulus was assumed as the tensile module of elasticity in the longitudinal direction (Table 3). At first, a fully rigid rotational behaviour was assigned to the joints. A monotonic increase in the load was simulated and the corresponding horizontal displacement of the point where the laser was placed in the experimental test was considered output data. The load–displacement response deriving from the numerical analysis, representing the theoretical behaviour with fully rigid joints, is represented as well in Figure 15. By comparing the experimental and numerical curves, a significant difference in the rigidity of the system was observed. The lower rigidity of the GFRP portal frame tested emphasised that the joints were characterised by a rotational stiffness that could not be modelled with a fully rigid behaviour. This aspect, confirming what was already shown by the beam-to-column specimens, showed that the joints were characterised by a semi-rigid behaviour.

The distribution of the displacement deriving from the numerical model with fully rigid joints is represented in Figure 17. Specifically, it represents the deformed shape concerning a lateral load of 45 kN. The experimental values of the displacement in correspondence with the LVDTs placed on the structure are shown as well. The comparison shows the significant discrepancy between the fully rigid model and the actual behaviour of the portal frame.

In order to have a more reliable numerical model, the rotational stiffness of the joints was calibrated with respect to the experimental results. As result, a numerical model characterised by the semi-rigid joints was implemented. The displacement distribution of the numerical model with semi-rigid joints is represented as well in Figure 17. Specifically, a rotational rigidity equal to 2.6 MNm/rad was assigned to the joints. This value of the rotational stiffness corresponds to the average value obtained from the first set of tests of the beam–column specimen. It can be adopted for the modelling of the system in view of modifications aimed at increasing its structural efficiency with respect to oversizing issues.

#### 4.2.3. Strain Distribution

The strain distribution over the left column was assessed by means of strain gauges placed both on the flange and on the lateral web.

The GFRP column cross-section strain distribution was assessed at two different heights, 1 m and 2.4 m, by means of six strain gauges placed on the lateral web of each section (Figure 6). For each section, the strain gauges, placed on the same horizontal line, record the axial deformation in the vertical direction. The strain distribution of the two monitored sections, corresponding to representative values of the load during the test, is represented in Figure 18, respectively, for the sections placed at a height of 2.4 m and 1 m.

The strain distribution of the section at 2.4 m confirmed that the section is subjected to simple bending, with the neutral axis in proximity to the geometric axis of the cross-section. This also emphasises a quite homogeneous response of the material with a similar behaviour in compression and in tension. The cross-section remained plane during the test, and a quite linear increase in the strain with the load was observed. The cross-section at 1.0 m was characterised by very low values of the strain, confirming this position to be in proximity to the point in which the moment is null, as shown by the numerical model response.

Strain gauges were also placed on the external flange of the column, at different heights along the same vertical line, and recording the axial vertical strains (Figure 6). The distribution of the longitudinal strains along the column’s external flange is shown in Figure 19. The strain distribution confirmed the point of null moment to be placed at a height of about 1.0 m. The flange resulted to be in compression for the points situated above this point, and in tension for the points below it. With respect to the compressed zone, the strain gauges recorded a quite-linear increase in the deformation with the applied load at both positions of 1.2 m and 2.4 m.

Within the zone in tension, the deformation distribution showed an inversion in its trend at a height of 0.75 m. As a matter of fact, the sections in proximity to the column base showed lower values of the strains. This experimental evidence is due to the presence of the connecting elements, L-shape steel plates and fasteners, that bear part of the load reducing the concentration of the stress in the column flange. The strain distribution of the strain along the column height confirms that the GFRP member, out of the joint, remains in elastic conditions during the entire test, confirming that plastic deformations are mainly concentrated in the connecting elements. This aspect is in line with the analysis of the failure mode that showed the shear failure of the screws without significant damage in the GFRP elements.

## 5. Conclusions

The study presents the results of an experimental study aimed at investigating the mechanical behaviour of GFRP frames including steel plate fastened joints. The main outcomes of the research are reported as follows:Beam-to-column specimens are characterised by a semirigid behaviour exhibiting a failure mode characterised by the shear rupture of the fasteners. Cyclic load tests show a proper response of both the specimens with a symmetrical hysteresis curve with energy dissipation values increasing with the load;The use of washers in the fasteners reduces the values of the residual displacements delaying the beginning of the plastic deformation of the joint;The analysis of the strain field during the test emphasises that the GFRP member out of the connection behaves in elastic conditions while plastic deformations are concentrated in the joint portion;The maximum capacity of the joint is much higher than the expected values from structural analysis. This aspect emphasises that the structure, mainly designed to guarantee practical logistic aspects related to the geometry of the prefabricated elements adopted in the modular prototype, is structurally oversized.GFRP frames laterally loaded exhibit a mechanical response characterised by the failure of the most stressed joints, consistent with the behaviour shown by the beam-to-column elements. The analysis of the strain distribution confirms the concentration of plastic deformations in the connecting elements. The localisation of the damages in the connecting elements, without affecting beams and columns, may represent an important aspect in view of repairing interventions in which the structural members may be reused;The comparison with numerical analysis emphasises the semi-rigid nature of the joints. The adoption of proper rotational stiffness of the joint allows exhaustive modelling of the mechanical behaviour of the frame that can be adopted in view of modifications aimed at increasing its structural efficiency with respect to oversizing issues.

All in all, the study shows that GFRP prefabricated elements represent a valuable technical solution for the implementation of modular prototypes, capable of exhibiting sufficient structural strength. The proposed connecting system, a practical system suitable for installation and disassembly processes, provides a semi-rigid behaviour of the joint with a concentration of damage in the connecting elements. However, the experimental results also emphasise that the exhibited capacity is much higher than the required strength, showing that there is still room for improvement to optimise the size of the connecting system in order to reduce oversizing issues. The simplified numerical model proposed in the study can be adopted as a tool to achieve such improvement. For further development, a joint-based design with nonlinear analysis could be done.

## Figures and Tables

**Figure 1 materials-15-08282-f001:**
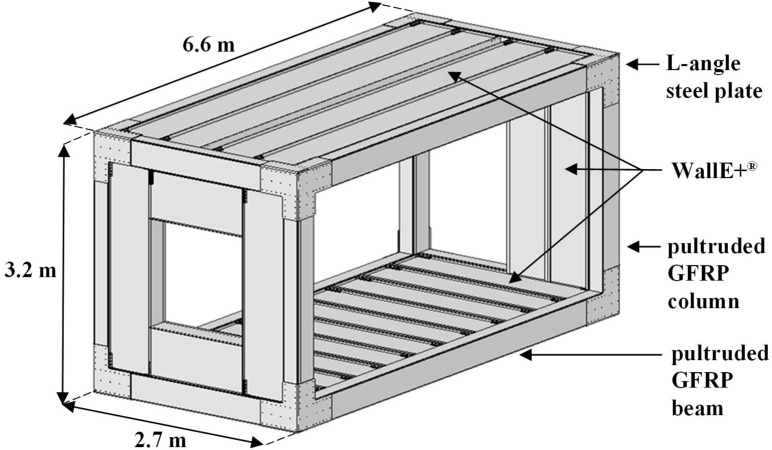
3D representation of the MOOVABAT modular element.

**Figure 2 materials-15-08282-f002:**
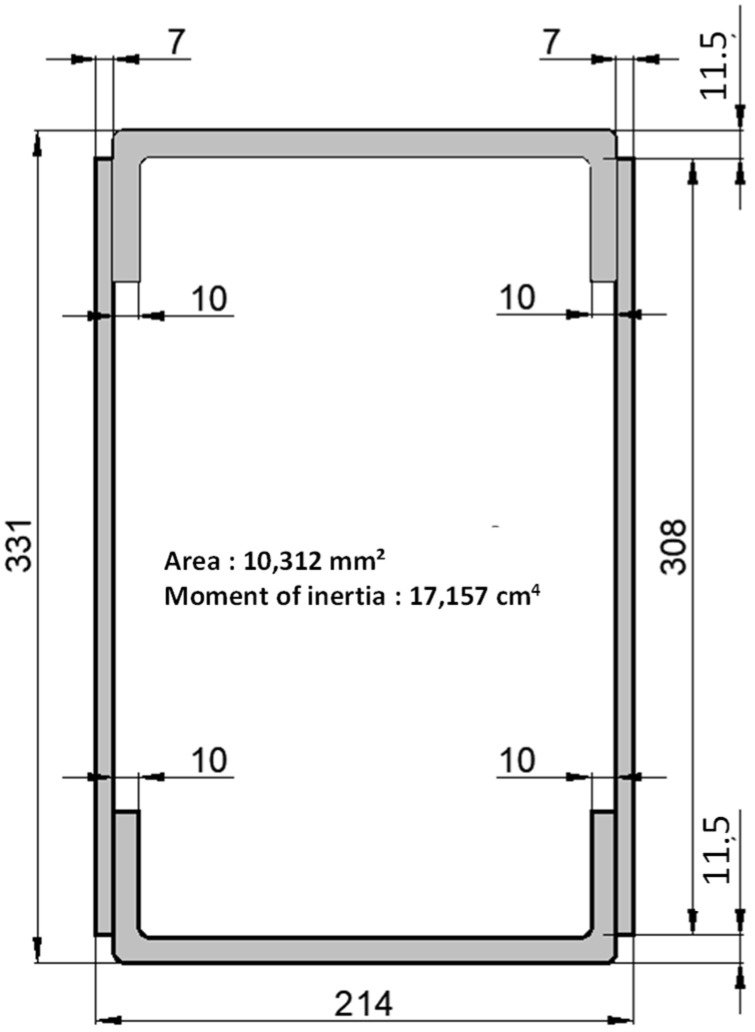
Beam and column GFRP cross section and geometric properties (measures in mm).

**Figure 3 materials-15-08282-f003:**
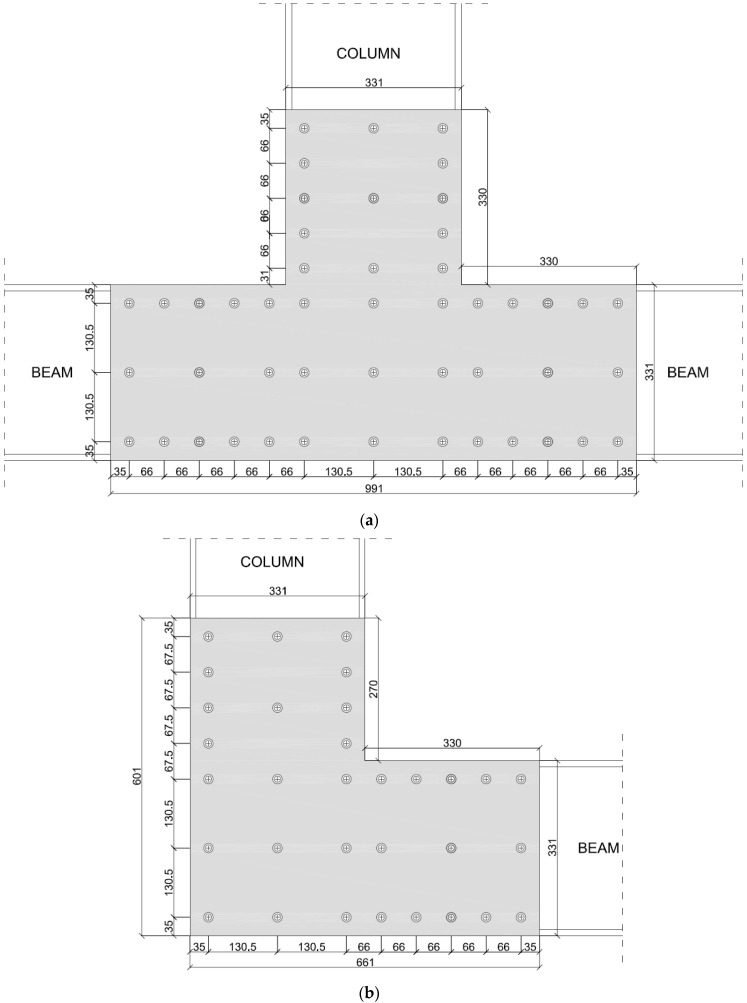
Close-up of the joints: (**a**) T-joint; (**b**) L-joint (measures in mm) (centre of cross signs represents the longitudinal axis of the screws).

**Figure 4 materials-15-08282-f004:**
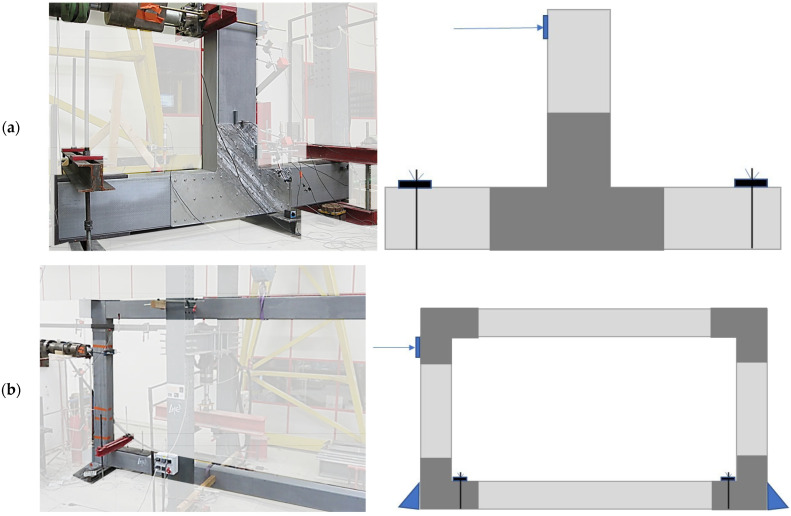
View of the specimen tested: (**a**) beam-to-column; (**b**) portal frame.

**Figure 5 materials-15-08282-f005:**
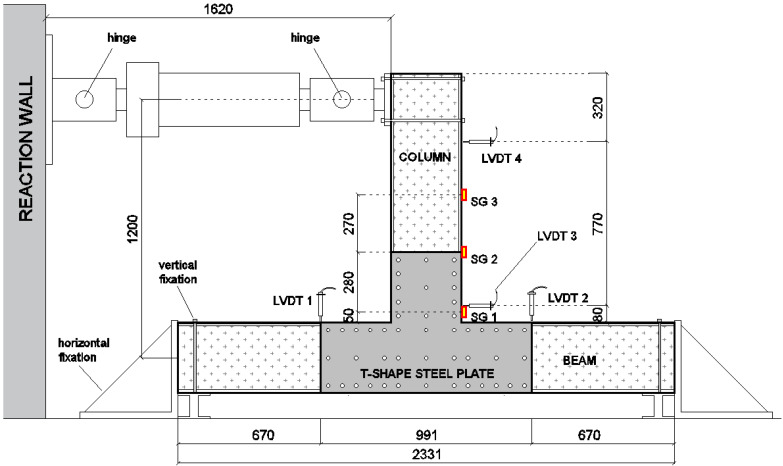
Test set-up of beam-to-column element (specimens T1 and T2_W) (measures in mm).

**Figure 6 materials-15-08282-f006:**
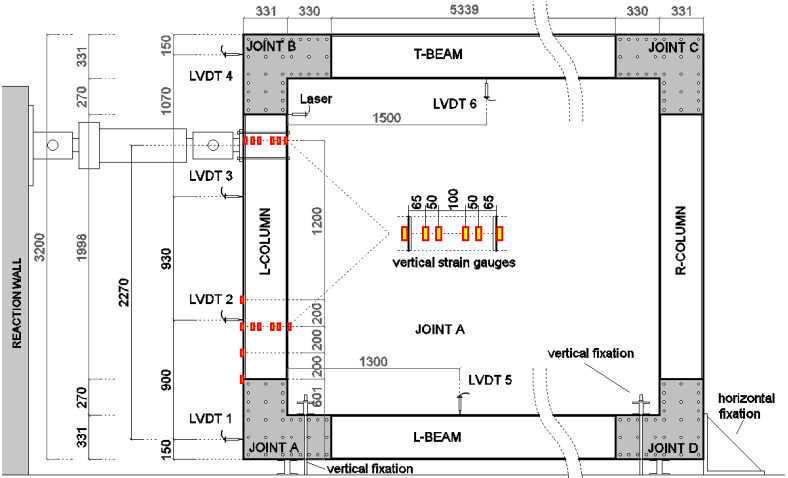
Test set-up of the portal frame, specimen PF (measures in mm).

**Figure 7 materials-15-08282-f007:**
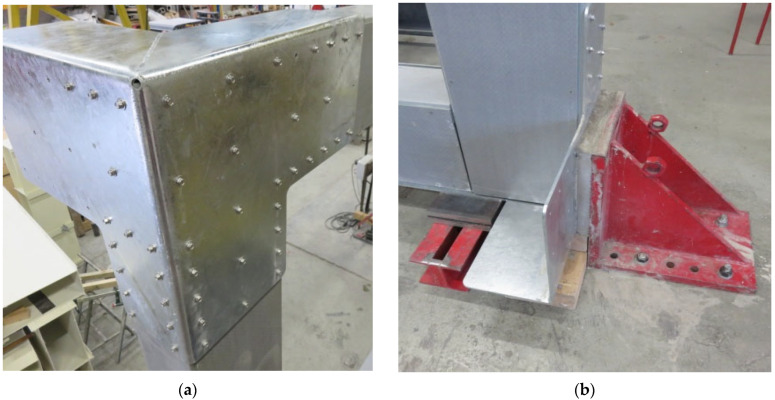
Close-up of a portal frame joint: (**a**) external side; (**b**) inner side.

**Figure 8 materials-15-08282-f008:**
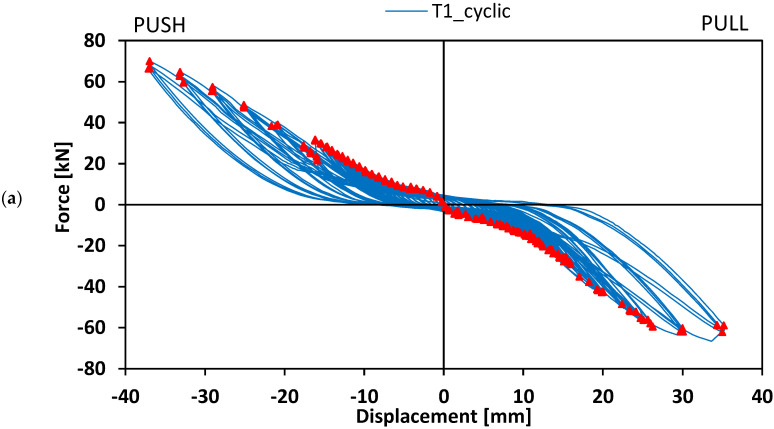
Load–displacement hysteresis curves concerning beam-to-column elements: (**a**) specimen T1; (**b**) specimen T2_W.

**Figure 9 materials-15-08282-f009:**
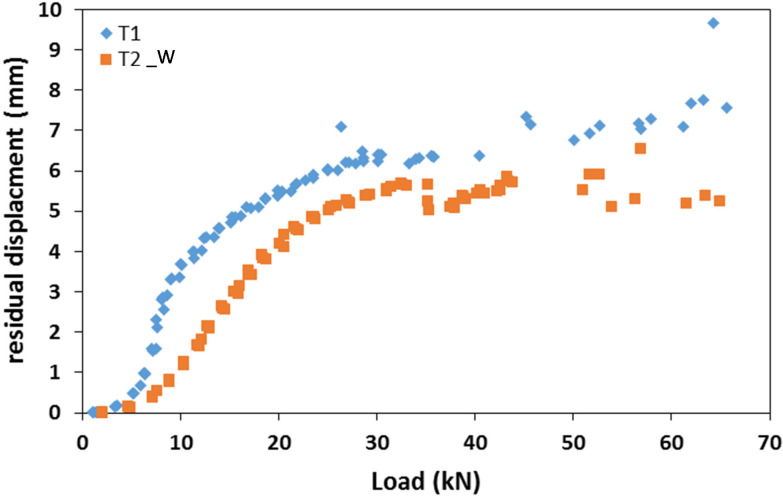
Residual displacement evolution during the cycling loading.

**Figure 10 materials-15-08282-f010:**
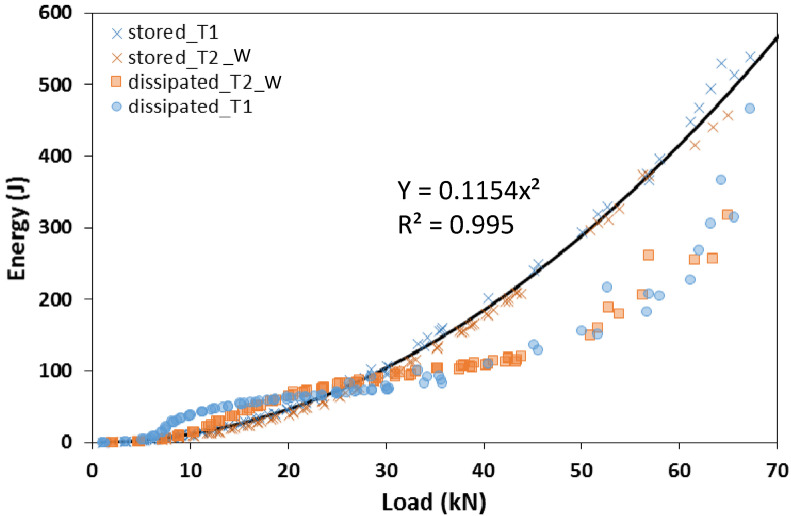
Stored and dissipated energy as a function of the maximum load for each cycle concerning T1 and T2_W specimens.

**Figure 11 materials-15-08282-f011:**
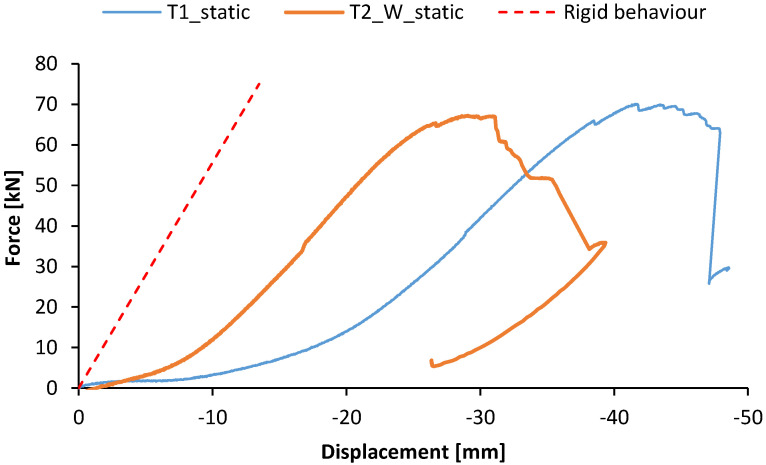
Load–displacement response of T1 and T2_W specimens tested in static loading conditions.

**Figure 12 materials-15-08282-f012:**
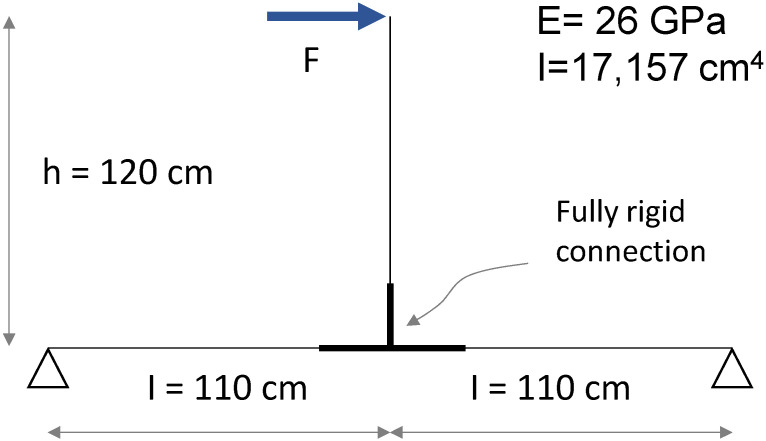
Structural scheme of the beam-to-column element adopted for the mechanical analysis.

**Figure 13 materials-15-08282-f013:**
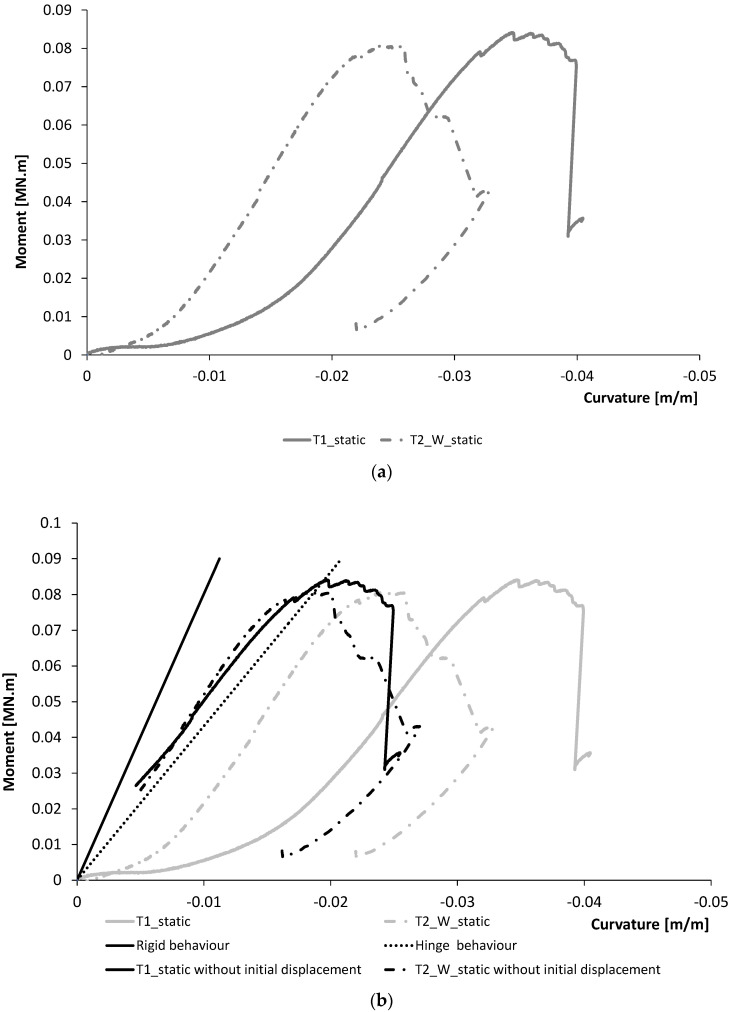
Moment–rotation response of T1 and T2_W specimens tested in static loading conditions. (**a**) Experimental results with sliding. (**b**) Experimental results without initial sliding and comparison with calculations.

**Figure 14 materials-15-08282-f014:**
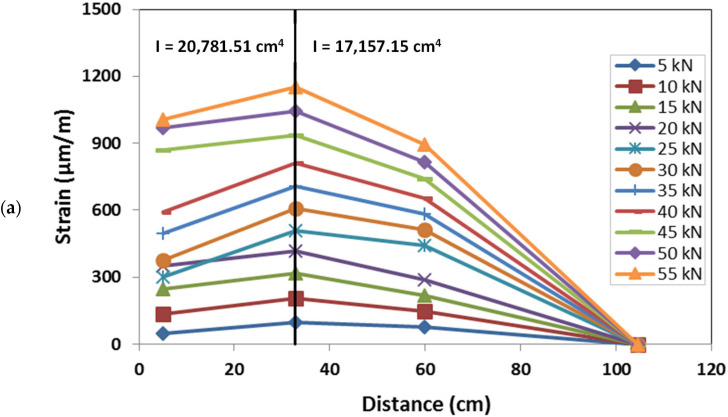
Strain distribution in the column flange at different loads: (**a**) pushing phase; (**b**) pulling phase.

**Figure 15 materials-15-08282-f015:**
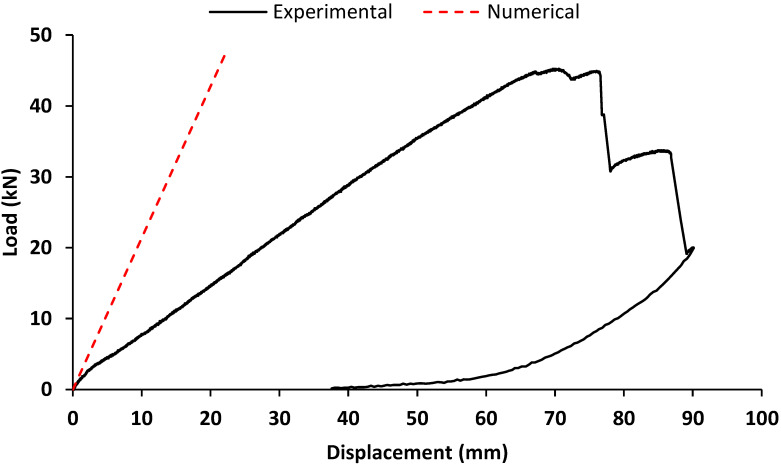
GFRP portal frame load–displacement behaviour: experimental and numerical curves.

**Figure 16 materials-15-08282-f016:**
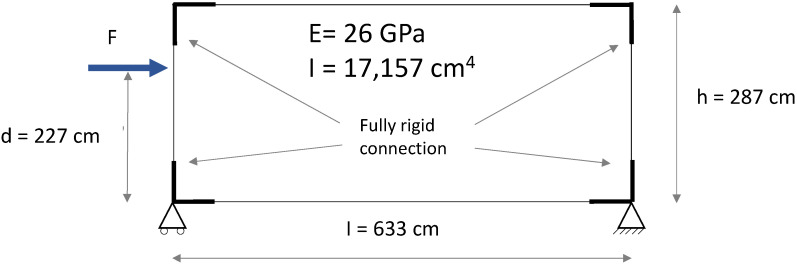
Structural scheme adopted in the numerical model of the GFRP portal frame.

**Figure 17 materials-15-08282-f017:**
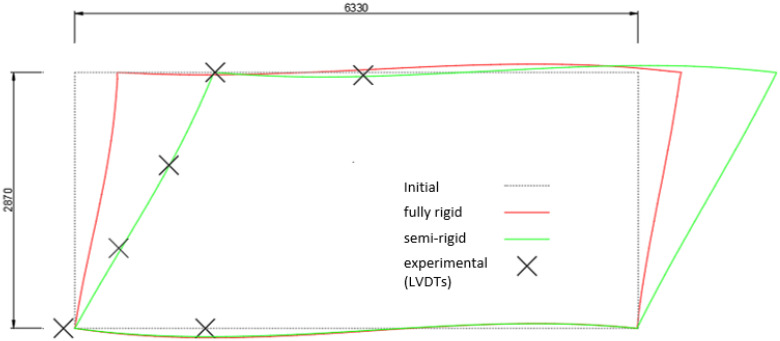
Deformed shape of the GFRP frame corresponding to a lateral load of 45 kN: experimental displacement values, numerical displacement distributions with fully rigid and semi-rigid behaviours (measures in mm).

**Figure 18 materials-15-08282-f018:**
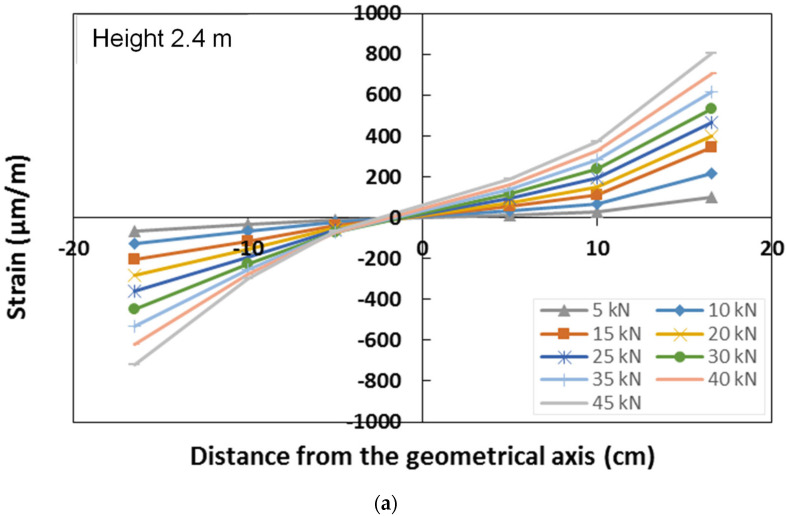
Strain distribution along the lateral web of the column: (**a**) section at a height of 2.4 m; (**b**) section at a height of 1.0 m.

**Figure 19 materials-15-08282-f019:**
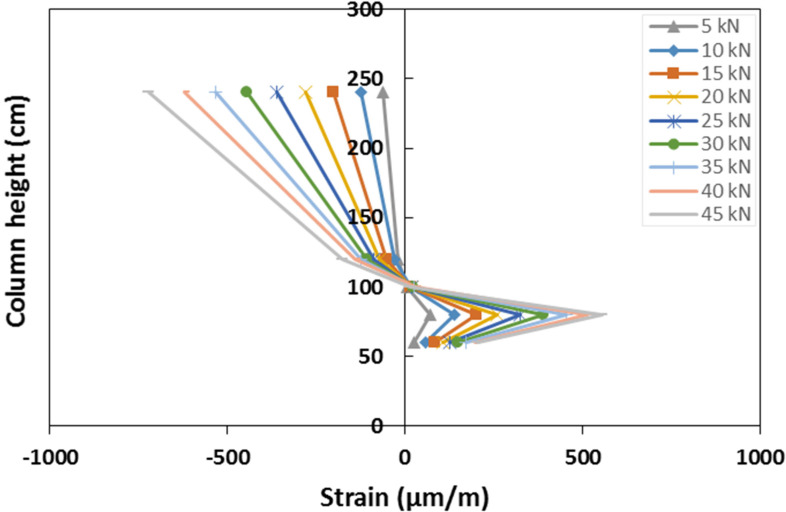
Strain distribution along the external flange of the column.

**Table 1 materials-15-08282-t001:** Loads considered in the preliminary structural analysis.

Type of Load	Value	Unit	Description
Permanent actions (*G*)	0.19	kN/m	deck, facade, and frame members
	0.92	kN/m^2^	deck concrete slab
	0.15	kN/m^2^	ventilation
	0.10	kN/m^2^	partitions
	0.36	kN	connection steel plate
Variable actions (*Q*)	5.00	kN/m^2^	-
Wind loads (*W*)	0.53	kN/m^2^	-
Snow load (*S*)	0.56	kN/m^2^	-
Accidental snow load (*AS*)	1.08	kN/m^2^	-

**Table 2 materials-15-08282-t002:** Load cases and maximum internal forces deriving from structural analysis.

Load Case	*N*	*V*	*M_b_*	*M_j_*
	[kN]	[kN]	[kNm]	[kNm]
^(^*^)^ 1.35·*G*	−8.2	3.3	5.7	−6.6
1.35·*G* or 1.35·*G* + 1.5·*Q*	3.9	1.2	2.2	2.5
1.35·*G* + 1.5·*S*	7.4	2.5	4.7	5.2
1.35·*G* + 1.5·*S* + 0.9·*W*	8.3	3.2	5.1	6.3
1·*G* + 1.5 *W*	−6.7	3.1	−3.1	−3.9
1.35·*G* + 1.5·*W* + 0.9·*S*	8.1	1.1	2.8	2.9

^(^*^)^ serviceability load case.

**Table 3 materials-15-08282-t003:** Main mechanical properties of pultruded GFRP elements.

Properties	Longitudinal	Transversal	Reference Standard
Tensile strength	400 MPa	30 MPa	ASTM D638
Tensile modulus of elasticity	26 GPa	8 GPa	
Compressive strength	220 MPa	70 MPa	ASTMD695
Compressive modulus of elasticity	18 GPa	7 GPa	
Shear strength	30 MPa	-	ASTM D2344
Shear modulus of elasticity	3 GPa	-	EN 13706
Poisson’s ratio	0.28	0.12	ASTM D3039

## Data Availability

Data available on demands.

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
