# Peer review of "Mechanical Characterisation of GFRP Frame and Beam-to-Column Joints Including Steel Plate Fastened Connections"

_materials, 2022, doi:10.3390/ma15238282_

Round 1
Reviewer 1 Report
The manuscript proposes an interesting study about the definition of an innovative composite structure with bolted joints. The experimental setup is rigorously described. The discussion of some results should be further improved as well as the description of the numerical analyses.
The page numbering is not working, notably page 6 of the pdf file is numbered as page 2. In the comments below the page numbering is referred to the pdf file.
Overall, the revised version of the manuscript should solve the following issues:
· Paragraph 1 - explain in this section the meaning of the acronyms SLS and ULS.
· Page 5 - report a figure with the simplified structural model considered in the analysis showing the position of the applied loads and constraints.
· Page 8 - considering the comment below Equation 2, it is mentioned the coordinate z_i but Equation 2 presents the coordinate y_i.
· Page 10 – it is reported that the static test is performed after the cyclic test. A remark about this aspect is that the cyclic test might have produced cracks or damage to the structure and therefore the results of the static test are altered by this aspect. This is justified by the comments in paragraph 4.1.1 about the plastic strain and the loss of stiffness of the joint curve shown in Figure 8.
· Figure 12 – what are the characteristics of the numerical model? Is it a finite element model? In this case, what kind of elements were utilized? Did you consider an elastic or elastic-plastic model more the materials? Give more details about the numerical model. Moreover, a screw joint provides further elastic contributions in a numerical model as well as a non-linear contribution related to friction. Neglecting these attributes gives your model poor accuracy, these limitations should be further discussed. The literature presents different techniques to model the bolted joints.
· Figure 17 – considering that the semi-rigid model was calibrated against the results of the GFRP portal frame it is not surprising that its results agree with the experimental data of the GFRP portal frame, this is a recursive verification. Try, at least, to compare the numerical model with the T-joint to have a more significative validation of the model.
· The introduction should discuss some references about the structural behavior of bolted connections and estimation of their stiffness, such as: https://doi.org/10.1016/j.compstruct.2020.112770; https://doi.org/10.1016/j.compositesb.2021.109378
· Carefully check the manuscript for typos and misspellings.
Author Response
The page numbering is not working, notably page 6 of the pdf file is numbered as page 2. In the comments below the page numbering is referred to the pdf file.
Thanks for this comment, modification done.
Overall, the revised version of the manuscript should solve the following issues:
- Paragraph 1 - explain in this section the meaning of the acronyms SLS and ULS.
Thanks for this comment, modification done.
- Page 5 - report a figure with the simplified structural model considered in the analysis showing the position of the applied loads and constraints.
An additional figure is proposed on figure 4.
- Page 8 - considering the comment below Equation 2, it is mentioned the coordinate z_i but Equation 2 presents the coordinate y_i
Thanks for this comment, modification done.
- Page 10 – it is reported that the static test is performed after the cyclic test. A remark about this aspect is that the cyclic test might have produced cracks or damage to the structure and therefore the results of the static test are altered by this aspect. This is justified by the comments in paragraph 4.1.1 about the plastic strain and the loss of stiffness of the joint curve shown in Figure 8.
The static test is the last loading applied on the frame until the failure, a sentence has been introduce to clarify.
The two beam-to-column specimens, namely T1 and T2_W, were subjected to cyclic loading (Cyclic Test), and then to a final last static loading conditions with the load monotonically increasing up to the failure (Static Test). The cyclic test is stop when enough damaged is identified in the structure.
- Figure 12 – what are the characteristics of the numerical model? Is it a finite element model? In this case, what kind of elements were utilized? Did you consider an elastic or elastic-plastic model more the materials? Give more details about the numerical model. Moreover, a screw joint provides further elastic contributions in a numerical model as well as a non-linear contribution related to friction. Neglecting these attributes gives your model poor accuracy, these limitations should be further discussed. The literature presents different techniques to model the bolted joints.
A sentence have been added to clarify, basic frame modelling is used. “A usual beam model is used to get force-displacement response.” The siffness used was the one of the columns/beams test.
- Figure 17 – considering that the semi-rigid model was calibrated against the results of the GFRP portal frame it is not surprising that its results agree with the experimental data of the GFRP portal frame, this is a recursive verification. Try, at least, to compare the numerical model with the T-joint to have a more significative validation of the model.
The semi regid model was calibrated on th column/beams test and then applied on the frame.
“Specifically, a rotational rigidity equal to 2.6 MNm/rad was assigned to the joints. This value of the rotational stiffness correspond to the average value obtained from the first set of test of beam-column specimen. It can be adopted for the modelling of the system in view of modifications aimed at increasing its structural efficiency with respect to oversizing issues. “
- The introduction should discuss some references about the structural behavior of bolted connections and estimation of their stiffness, such as: https://doi.org/10.1016/j.compstruct.2020.112770; https://doi.org/10.1016/j.compositesb.2021.109378
The two reference have added in the text and a comment add, thank for this valuable comment, further work on modelling will be done in future based on these results.
“The aim of this is in a first approach to identify the boundary conditions of the connection. Further deeper mechanical modelling could be done to include the slipping effect of the connection which not the aim at this stage but FEM or non-linear analysis may be done [35, 36].”

Reviewer 2 Report
Dear authors,
It is an interesting and meaningful job with plenty of achievements from experimental, numerical and analytical formula results. My common comments were attached in the file.
Besides, is this beam-column connection treated as semi-rigid joint? And the hysteretic curve is not saturated enough and I think these two kind of joints may be not perfect enough. It can be improved in further researches.
Beset regards,
Reviewer

Author Response
Thanks a lot for this valuable review, all comments have been consider
- aimed have been replaced by aiming
- ULS and SLS have been fully detailed on the first page
- 3 new reference have been added to the document for modular construction
- the support limit correspond to the one that can be used in timber construction
- the loading constions of the frame has been claryfied, first cyclic loading and a last static test untill the failure
- figure 13 has been improoved to be more clear

Round 2
Reviewer 1 Report
The suggested review issues have been solved.
As far as I am concerned, the paper can be accepted for publication.
Reviewer 2 Report
accepted